# The E3 Ubiquitin Ligase ATL9 Affects Expression of Defense Related Genes, Cell Death and Callose Deposition in Response to Fungal Infection

**DOI:** 10.3390/pathogens11010068

**Published:** 2022-01-05

**Authors:** Tingwei Guo, Feng Kong, Carter Burton, Steven Scaglione, Blake Beagles, Justin Ray, Katrina M. Ramonell

**Affiliations:** 1Department of Biological Sciences, The University of Alabama, Tuscaloosa, AL 35401, USA; tingweig@usc.edu (T.G.); fkong@crimson.ua.edu (F.K.); cdburton1@crimson.ua.edu (C.B.); steven.r.scaglione@vanderbilt.edu (S.S.); abbeagles@crimson.ua.edu (B.B.); jbray5@crimson.ua.edu (J.R.); 2Center for Craniofacial Molecular Biology, University of Southern California, Los Angeles, CA 90089, USA

**Keywords:** plant defense, ubiquitination, fungal infection, plant-pathogen interaction, signal transduction pathways, E3 ubiquitin ligase, *Arabidopsis thaliana*

## Abstract

Plants use diverse strategies to defend themselves from biotic stresses in nature, which include the activation of defense gene expression and a variety of signal transduction pathways. Previous studies have shown that protein ubiquitination plays a critical role in plant defense responses, however the details of its function remain unclear. Our previous work has shown that increasing expression levels of *ATL9*, an E3 ubiquitin ligase in *Arabidopsis thaliana,* increased resistance to infection by the fungal pathogen, *Golovinomyces cichoracearum*. In this study, we demonstrate that the defense-related proteins PDF1.2, PCC1 and FBS1 directly interact with ATL9 and are targeted for degradation to the proteasome by ATL9. The expression levels of *PDF1.2*, *PCC1* and *FBS1* are decreased in T-DNA insertional mutants of *atl9* and T-DNA insertional mutants of *pdf1.2*, *pcc1* and *fbs1* are more susceptible to fungal infection. In addition, callose is more heavily deposited at infection sites in the mutants of *atl9*, *fbs1*, *pcc1* and *pdf1.2*. Overexpression of *ATL9* and of mutants in *fbs1*, *pcc1* and *pdf1.2* showed increased levels of cell death during infection. Together these results indicate that ubiquitination, cell death and callose deposition may work together to enhance defense responses to fungal pathogens.

## 1. Introduction

Plants face a variety of fungal pathogens in nature, which pose significant threats to successful growth and reproduction [1]. Unlike animals, plants lack a sophisticated adaptive immune system. Instead, plants have evolved diverse strategies to recognize pathogens and defend against them [1]. Our current view of the plant immune system can be represented as a ‘zigzag’ model with two layers [2]. The first layer of the immune system involves the recognition of conserved molecules from pathogens (pathogen-associated or microbe-associated molecular patterns, PAMPs or MAMPs), by plant transmembrane pattern recognition receptors (PRRs) that induces pattern-triggered immunity (PTI) [3]. However, pathogens that successfully colonize host plants can deliver virulence factors (effectors) into plant cells to counteract the effects of plant PTI, referred to as effector-triggered susceptibility (ETS) [2]. In this case, the second layer of the plant immune system comes into play which involves the recognition of effectors by plant nucleotide-binding leucine-rich repeat (NB-LRR) proteins, resulting in effector-triggered immunity (ETI) [4]. Both PTI and ETI activate complex immune responses, including the generation of reactive oxygen species, induction of phosphorylation cascades, the hypersensitive response (HR), the deposition of callose, and the production of antimicrobial compounds and defense hormones [5,6]. 

Such complex immune responses require tight regulation and coordination. The Ubiquitin/26S proteasome system plays an important role in degrading cellular proteins and maintaining protein homeostasis in both plants and animals. In recent years it has also emerged as one of the major regulatory signals in plant-fungal interactions [7]. The covalent linkage of ubiquitin, a 76-amino-acid-long protein, to a target protein is a key step in protein degradation via the ubiquitin proteasome system. In general, protein ubiquitination is mediated by three essential enzymes: ubiquitin-activating enzymes (E1), ubiquitin-conjugating enzymes (E2), and ubiquitin ligases (E3). E3 ligases are responsible for recognizing and ubiquitinating specific proteins, thus targeting them for degradation to the proteasome [8]. In *Arabidopsis* there are more than 1300 genes that are predicted to encode E3 ligases while E1 and E2 enzymes are encoded by only 2 and 37 genes, respectively [9]. The large number and diversity of E3 ligases allow for response specificity in different signaling pathways. Current research classifies plant E3 ligases into four main categories based on their structural features and mechanism of action: HECT (Homologous to E6-associated protein C-Terminus), RING (Really Interesting New Gene), U-Box, and CRL (Cullin-RING Ligases) [10]. Although many studies have shown that ubiquitin ligases play a critical role in plant defense, the detailed functions and regulation of these enzymes remains unclear. 

In *Arabidopsis thaliana* the ATL9 protein is an E3 ubiquitin ligase that is induced by chitin and involved in basal resistance to the biotrophic fungal pathogen, *Golovinomyces cichoracearum,* the causal agent of powdery mildew disease [11,12]. However, little is known regarding the detailed function and regulation of ATL9 in responses against fungal pathogens. Our previous data showed that the expression of three genes, *PCC1*, *PDF1.2*, and *FBS1* were regulated by the expression of *ATL9* in response to pathogen infection [11]. *F-Box stress-induced 1* (*FBS1*) is an F-box protein that is induced in response to wounding stress, osmotic stress, methyl jasmonate (MeJA), salicylic acid (SA), and abscisic acid (ABA) treatments [13]. *Pathogen and circadian controlled 1* (*PCC1*), is a pathogen-responsive gene that is regulated by the circadian clock [14]. *PCC1* is involved in the regulation of flowering time in *Arabidopsis* [14,15]. In addition, *PCC1* has also been demonstrated to operate in defense responses against the hemi-biotrophic oomycete pathogen *Phytophthora brassicas* and the necrotrophic fungal pathogen *Botrytis cinerea* [14]. *Plant defensin 1.2 (PDF1.2*), has a myriad of functions, including response to jasmonic acid (JA) [16,17], ethylene [16], insects [18], and defense against fungal pathogens [17,18]. 

In this study, we have expanded on our previous work, focusing on ATL9′s interacting proteins and the consequences of ATL9 overexpression [11]. Our data show that PDF1.2, PCC1 and FBS1 directly interact with and are ubiquitinated by ATL9. An in vivo ubiquitination assay showed that ATL9 can target these three proteins for degradation via the 26S proteasome. Furthermore, *atl9* T-DNA insertional mutants showed reduced expression of *PDF1.2*, *PCC1*, and *FBS1*, impairing the plant’s resistance to powdery mildew. We also observed that cell death typically induced during the hypersensitive response is also associated with increased *ATL9* expression levels. Interestingly, our results also show that *ATL9* overexpression inhibits callose deposition in infected plants (Figure 7). Callose deposition is a strategy used by plants to reinforce the cell wall against penetration by fungal hyphae during infection [19,20]. Based on these data, we propose a model of the role that ATL9 may play during early infection in *Arabidopsis* and a potential mechanism demonstrating how ubiquitination, cell death and callose work together to enhance the defense response against *G. cichoracearum.*

## 2. Results

### 2.1. Mutants of pdf1.2, pcc1 and fbs1 Are More Susceptible to G. cichoracearum Infection

To confirm if *PDF1.2*, *PCC1* and *FBS1* are involved in defense against fungal infection, T-DNA insertional mutants of *pdf1.2, pcc1* and *fbs1* were infected with powdery mildew and their phenotype was evaluated. Six days post inoculation (dpi), leaves of *atl9* (At2g35000), *pdf1.2* (AT5G44420), *pcc1* (AT3G22231), and *fbs1* (AT1G61340) all had more extensive fungal growth and the fungi had generated more spores when compared to Columbia wild type (*Col-0*), demonstrating that these mutants are more susceptible to powdery mildew infection (Figure 1). *NahG,* a transgenic plant deficient in salicylic acid (SA) production, was used as a positive control and displayed the expected hypersusceptible phenotype when infected with powdery mildew. Taken together these results indicate that *FBS1*, *PCC1* and *PDF1.2* may play a critical role in defense responses against powdery mildew.

### 2.2. ATL9 Directly Interacts with PDF1.2, FBS1, and PCC1

Since mutations in *PDF1.2*, *PCC1*, and *FBS1* result in a more susceptible phenotype when challenged with powdery mildew (Figure 1), we hypothesized that the above genes may encode interacting partners of ATL9. To confirm this, we conducted two experiments to investigate protein-protein interactions [21,22]. In the first set of experiments, we used the yeast two-hybrid (Y2H) assay to screen for possible interactions. A combination of bait plasmid pGBKT7-ATL9 and prey plasmids containing pGADT7-PDF1.2, pGADT7-PCC1, and pGADT7-FBS1 were transformed into yeast strains and then mated for 12 h. SD/-Ade/-His/-Leu/-Trp/X-α-gal plates were used to indicate a positive protein-protein interaction. As shown in Figure 2, mating cultures grew on selection media and had positive β-gal activity when the ATL9 bait (pGBKT7-ATL9) was combined with PDF1.2, PCC1, and FBS1 in the prey vector. As a control, the bait plasmid pGBKT7-ATL9 was mated with empty prey vector pGADT7 and no β-gal activity was observed (Figure 2). These data indicate that PDF1.2, PCC1 and FBS1 were strong candidates for potential interacting partners of ATL9.

To further investigate and confirm the interaction between ATL9 and its targets, the in vivo interaction observed in the Y2H assay was verified using a bimolecular fluorescence complementation assay (BiFC) [21,22]. ATL9 was fused with the C-terminal portion (amino acids 156–239) of yellow fluorescent protein (YFP; pDEST-VYCE-ATL9) and PDF1.2, PCC1, and FBS1 were fused with the N-terminal portion (amino acids 1–173) of YFP (pDEST-VYNE-PDF1.2, pDEST-VYNE-PCC1, and pDEST-VYNE-FBS1) and then co-infiltrated into *Nicotiana benthamiana* leaves. YFP fluorescence was imaged after 36 h using confocal laser-scanning microscopy (Figure 3). Results showed that YFP fluorescence was detected in cells co-infiltrated with: pDEST-VYCE-ATL9/pDEST-VYNE-PDF1.2, pDEST-VYCE-ATL9/pDEST-VYNE-PCC1, and pDEST-VYCE-ATL9/pDEST-VYNE-FBS1 reconfirming the results observed in the Y2H assays (Figure 2). No YFP fluorescence was detected in tobacco leaves co-infiltrated with only one of the constructs or with any combinations of the empty vectors (Appendix A). Together these results demonstrate that ATL9 can directly interact with PDF1.2, PCC1 and FBS1 in planta. 

### 2.3. ATL9 Is Required for Degradation of PDF1.2, FBS1, and PCC1 In Vivo

The interactions between ATL9 and PDF1.2, FBS1, and PCC1 shown above suggest that ATL9 is responsible for ubiquitination of these proteins and that they are degraded via the 26S proteasome. To verify this hypothesis, we fused PDF1.2, FBS1, and PCC1 to the green fluorescent protein (PDF1.2:GFP, FBS1:GPF, and PCC1:GFP) and co-infiltrated them with a vector overexpressing *ATL9* (35S:*ATL9*) in tobacco leaves. Thirty-six hours post infiltration, the transformed tobacco leaves were treated with either the protein-synthesis inhibitor cycloheximide (CHX) or a combination of CHX and MG132, a proteasome inhibitor. Fluorescent emission in the treated leaves was then monitored by confocal laser-scanning microscopy. The results are shown in Figure 4 and the white arrows indicate the areas where the protein is significantly degraded. The GFP signal fades gradually if PDF1.2:GFP has been co-transformed with 35S:*ATL9* (Figure 4A, column I). However, if samples have been treated with CHX/MG132, the progression of the degradation is much slower, suggesting that PDF1.2 is degraded via the 26S proteasome (Figure 4A, column II). Fluorescent emission of PDF1.2:GFP only or PDF1.2:GFP combined with an empty vector shows little to no change in degradation of the signal (Figure 4A, column III and IV). PCC1:GFP and FBS1:GFP show similar results to PDF1.2:GFP when co-transformed with 35S:*ATL9* (Figure 4B,C). We also evaluated the influence of the individual vectors used in this experiment (Appendix A) and none of them caused any change in degradation. 

To further verify that degradation of the three genes is caused by ATL9 alone, total protein was extracted from co-infiltrated tobacco leaves at 0 and 6 h followed by immunoblotting using an anti-GFP antiserum. The results show that the quantity of PDF1.2-GFP, PCC1-GFP, and FBS1-GFP reduces when co-infiltrated with 35S:ATL9 (Figure 5A). While PDF1.2-GFP, PCC1-GFP, and FBS1-GFP remain unchanged without co-infiltration with ATL9 (Figure 5B). Together these findings strongly suggest that in vivo degradation of FBS1, PCC1 and PDF1.2 is mediated by ATL9 and the proteasome.

### 2.4. Expression Pattern of Target Genes during Plant Defense Response

Since *atl9*, *pdf1.2*, *fbs1* and *pcc1* mutants are more susceptible to *G. cichoracearum* than wild-type plants and direct interaction has been detected between ATL9 and these proteins (Figure 1, Figure 2 and Figure 3), we were interested in investigating the expression patterns of these genes during fungal infection. *Arabidopsis* plants were inoculated with *G. cichoracearum* and leaf samples were collected at different time points after infection for RNA extraction. Quantitative analysis of the gene expression patterns reveals that, compared to *Col-0* wild-type plants (Figure 6A), the relative expression level of *PDF1.2*, *FBS1*, and *PCC1* are largely repressed in the *atl9* mutant during infection (Figure 6B). In wild-type plants, *PDF1.2* shows consistent increased expression levels across all time points sampled during infection and is highly expressed at the twenty-four hour time point, as shown in Figure 5A. *FBS1* expression is highly induced at early time points (1 h and 30 min) and at the 24 h time point while *PCC1* expression was also highly induced at early time points (1.5 h and 2 h) and at the 16 h time point. However, expression levels of all three genes showed altered patterns in the *atl9* mutant compared with Col-0 wild-type (WT) during fungal infection. As shown in Figure 6B, *FBS1* was induced at early time points (1.5 h and 2 h), but expression of *FBS1* was repressed 4 h post-fungal infection. Interestingly *PCC1* expression was decreased across most of the time points during infection. In addition, *PDF1.2* expression was significantly decreased in the *atl9* mutant compared to Col-0 during fungal infection. However, *PDF1.2* expression was induced at later time points (4 h, 16 h, and 24 h) in the *atl9* mutant after fungal infection. Together these results indicate that *ATL9* is influencing the up regulation of *PDF1.2* and *PCC1* during defense responses against powdery mildew and it is important for the increased expression levels of *FBS1* observed at early time points after the initial infection. 

### 2.5. Deposition of Callose during Fungal Infection

Callose is a polysaccharide that plays an important role in a variety of processes during plant development and in defense responses [23]. Previous research has shown that callose is deposited in the leaf early in the infection process and can efficiently prevent fungal penetration [24]. Furthermore, callose and callose synthase have been identified as negative regulators of the SA defense pathway [25]. SA can mediate a diverse set of defense genes, is involved in cell death responses, and has significant cross talk with both the JA and ethylene defense signaling pathways [26]. Given that FBS1, PCC1, and PDF1.2 are known to be involved in SA, JA, and ethylene signaling and are targeted for degradation by ATL9, we wanted to determine if callose deposition was altered in mutants of any of these four genes [27,28,29].

*Arabidopsis* plants were infected with powdery mildew and leaf tissue was collected at four different time points after infection (3 h, 6 h, 12 h and 5 days). Callose deposition was determined by aniline blue staining and leaf samples were observed using confocal microscopy. The images of callose staining of leaves after infection for 3, 6 and 12 h or 5 days is shown in Figure 7A–D, and a quantitative assessment of the number of callose deposits per mm^2^ leaf tissue is shown in Figure 7E. The results showed that there were different patterns of callose deposition depending on the mutant and timepoint that were being observed. After 3 h of infection, more callose deposition was observed in mutants of *atl9*, *fbs1*, *pcc1*, and *pdf1.2* compared to *Col-0* controls, while very little callose deposition was detected in 35S:*ATL9*, as shown in Figure 7A. Six hours post infection, callose deposition largely increases in all mutants tested, but more callose deposition can be observed in *Col-0*. Meanwhile, at the 6 h timepoint there is no callose observed in the 35S:*ATL9* overexpression line (Figure 7B). Callose deposition continues to increase in *Col-0* and *atl9* 12 hours-post-infection and a small amount of callose deposition was observed in the 35S:*ATL9* line at the 12 h time point (Figure 7C). In contrast, callose deposition declines at the 12-hour time point in the *fbs1*, *pcc1* and *pdf1.2* mutants relative to *Col-0* (Figure 7C). We also examined samples 5 days post infection, in order to evaluate callose deposition at a late time point in the infection progression. As shown in Figure 7D, the *ATL9* overexpression mutant has more callose deposition compared to *Col-0*, while callose deposition has been largely repressed in the *atl9, fbs1, pdf1.2 and pcc1* mutant lines in response to fungal infection. 

In summary, mutants of *pdf1.2, fbs1 and pcc1* have more callose deposition at early stages of infection (6 h) while the overexpression line of *ATL9* displays more callose deposition during the late stages of infection (5 days). Callose deposition in mutants of *atl9, fbs1, pcc1 and pdf1.2* largely decreases at later time points during infection and is much less than that seen in wildtype control leaves. In addition, callose deposition in the 35S:*ATL9* over-expression line was highly induced. These results indicate that mutants of *atl9*, *fbs1*, *pcc1* and *pdf1.2* can synthesize and accumulate more callose at earlier time points after infection with powdery mildew than *Col-0* and 35S:*ATL9* over-expression plants. Callose deposition in 35S:*ATL9* plants is apparently delayed, suggesting that less callose deposition at the points of hyphal penetration may facilitate the delivery of other plant defense compounds during early infection stages.

### 2.6. Cell Death during Fungal Infection

Previous studies showed that *FBS1, PCC1*, and *PDF1.2* are involved in the three major defense pathways mediated by SA, JA and ethylene [14,27]. The expression levels of *FBS1*, *PCC1*, and *PDF1.2* are reduced in the *atl9* mutant during fungal infection, suggesting that *ATL9* might be involved in the regulation of these hormonal pathways via interaction with *FBS1*, *PCC1*, and *PDF1.2*. Additionally, previous research in our lab has shown that induction of *ATL9* is dependent on the production of reactive oxygen species (ROS) [11]. Recent studies have indicated that treating plants with SA and JA will induce cell death [30] and that increasing ROS will elevate cell death [31]. Host cell death is known to be an important response used to limit nutrient supply to biotrophic pathogens such as powdery mildew during infection [3,4,6]. Therefore, we hypothesize that cell death may be involved in and associated with expression of *ATL9* during the early stages of infection. To verify this prediction, leaves were infected with powdery mildew for 5 days and then stained with trypan blue to observe cell death (Figure 8). Results showed that 35S:*ATL9* plants displayed a higher level of trypan blue staining which equates to more cell death in the leaves of the *ATL9* overexpression plants. Meanwhile, signs of cell death in *atl9* loss of function mutants were lower than that observed in *fbs1*, *pcc1* and *pdf1.2* and more similar to the pattern observed in Col-0 wild-type. In addition, the *fbs1*, *pcc1* and *pdf1.2* mutants showed increased cell death compared to Col-0 but less than what was observed in 35S:*ATL9* plants (Figure 8). Collectively, these results indicate that increased expression levels of *ATL9* do stimulate cell death and decreased or no expression of *FBS1*, *PCC1* and *PDF1.2* will also result in increased cell death in *Arabidopsis*. In summary, we predict that during early infection induction of ATL9 is responsible for targeting FBS1, PCC1, and PDF1.2 for degradation to the 26S proteasome to promote cell death in the leaf epidermal cells that are points of hyphal penetration to limit fungal infection. 

## 3. Discussion

In the current work we have expanded on our previous studies, focusing on proteins that interact with the E3 ligase ATL9. We present evidence, via bimolecular fluorescence complementation assay and yeast-two hybrid assay, that the proteins PDF1.2, PCC1 and FBS1 directly interact with ATL9 (Figure 3, Figure 4, Figure 5 and Figure 6). Our previous work showed that ATL9 is localized to the endoplasmic reticulum [11]. PDF1.2 was found to localize to the ER bodies [32], where it can be secreted into the apoplastic space after a fungal infection [33]. PCC1 is localized to the plasma membrane [29] and FBS1 is predicted to localize to the mitochondria based on Y2H co-expression studies [34]. A future avenue of study would be to investigate the protein ubiquitination patterns of ATL9 and to determine the precise amino acid location of ubiquitination in FBS1, PCC1 and PDF1.2. The detailed roles of *PCC1, PDF1.2,* and *FBS1* in powdery mildew infection still largely remain unknown. Experiments involving multiple mutants of *pdf1.2, pcc1* and *fbs1* and overexpression lines of each gene are needed to provide more comprehensive information on the precise role of *ATL9, PCC1, PDF1.2,* and *FBS1* in immune responses against fungal pathogens. 

The SA, JA and ethylene defense pathways are well-defined but the points of crosstalk between them during infection are still unclear. Crosstalk between different hormonal defense pathways is hypothesized to reduce allocation costs by repression of unnecessary defenses [35]. For example, signaling crosstalk between SA and JA commonly shows reciprocal antagonism [36]. Research has shown that SA, JA, and ethylene are all positive regulators of cell death in plants and that targeted cell death (the hypersensitive response) is used as a mechanism to minimize infection at limited sites in the leaf [26,37,38,39]. FBS1, PCC1, and PDF1.2 are known to be involved in all three major defense pathways [14,27,40] and our data suggest that *ATL9* could be involved in adjusting SA, JA and ethylene responses through its interactions with *FBS1, PCC1, and PDF1.2*. However, further evidence is needed to definitively show the connection between these proteins and SA-, JA-, and ethylene-mediated signaling during fungal infection. 

Our data show that overexpression of *ATL9* confers resistance to powdery mildew [11,12]. Expression of *ATL9*, *FBS1*, *PCC1*, and *PDF1.2* has also been shown to be important for defense responses during fungal infection, and our current data show that ATL9 can directly bind to FBS1, PCC1 and PDF1.2 (Figure 2 and Figure 3) and target them for degradation to the proteasome (Figure 4 and Figure 5). In the current study, *ATL9* expression was induced only at early time points after fungal infection (Figure 6A). In contrast, *PDF1.2* expression was induced across all time points sampled during infection, *FBS1* expression was highly induced at the early time points (1 h and 30 min) and at the latest time point (24 h), while *PCC1* expression was also highly induced at the early time points (1.5 hand 2 h) and at the 16 h time point (Figure 6A). In the *atl9* loss-of-function mutant, expression levels of all genes showed altered expression patterns compared to Col-0 wild type during fungal infection, except for *PCC1* (Figure 6B). Considering all these data, we hypothesize that expression of *ATL9*, *PDF1.2, PCC1*, and *FBS1* during fungal infection may have complex regulation mechanisms and that *PDF1.2, PCC1*, and *FBS1* expression might also be regulated by other signaling pathways (or molecules) during infection. Moreover, our previous study showed that *ATL9* is also a short-lived protein within plant cells and it is degraded via the ubiquitin 26S proteasome pathway [12]. Together, this suggests that ubiquitination of PDF1.2, PCC1 and FBS1 proteins by ATL9 is likely not a continuous process during the plant defense response [41,42]. Further results show that fungal infection triggers increased expression of *ATL9*, *FBS1*, *PCC1* and *PDF1.2* and that *atl9* loss-of-function mutants show transcriptional repression of these three genes during infection. These results indicate that ATL9 plays a role in the upregulation of these genes at least at early timepoints during fungal infection. It appears that degradation of FBS1, PCC1, and PDF1.2 influences other aspects of the plant defense response. Since all genes tested were highly induced at early time points post infection, further studies that focus on exploring other known early-activated defense genes and pathways that may be affected by or interact with ATL9, PDF1.2, PCC1 and FBS1 will allow us to construct a more detailed picture of the intersecting early responses being coordinated against fungal infection in plants. 

Since *ATL9* is largely induced at early time points after infection, we also examined callose deposition in *atl9*, *pdf1.2*, *fbs1* and *pcc1* during early stages of infection. Our results show that mutants of *atl9, pdf1.2, fbs1 and pcc1* have more callose deposition early in infection, while callose deposition was largely repressed in 35S:*ATL9* overexpression lines in response to fungal infection. This suggests that *ATL9, FBS1, PCC1,* and *PDF1.2* might negatively regulate callose deposition during early infection stages. However, callose deposition was significantly increased in the overexpression line of *ATL9* at late timepoints after infection (16 h and 24 h) and was decreased in the mutant lines. Callose deposition during infection would block successful fungal penetration and colonization of the leaf tissue and our results showed that *atl9, fbs1, pcc1 and pdf1.2* mutants were more susceptible to powdery mildew (Figure 1). Thus, it seems likely that *ATL9, FBS1, PCC1,* and *PDF1.2* play a more complex role in the regulation of callose deposition in response to fungal infection. Especially considering our findings that *ATL9, FBS1, PCC1 and PDF1.2* are all highly induced 1.5 h after infection (Figure 6A) but that their expression levels are significantly decreased at the two hour timepoint (with the exception of *PDF1.2)* and remain repressed through the 24 h timepoint. Further experiments would be necessary to investigate the precise timing of ATL9 ubiquitination of these proteins to show that FBS1, PCC1 and PDF1.2 are influencing callose deposition.

Taking the data in the current study into consideration, we propose a putative model for ATL9′s function in defense responses against fungi and its possible interaction with other major defense pathways (Figure 9). 

During early infection stages, *ATL9* is highly induced after the plant is exposed to the pathogen and the classical defense pathways (SA, JA, ethylene) are activated. *FBS1*, *PCC1*, and *PDF1.2* are also upregulated (Figure 6A). Since ATL9 can bind and degrade all three of these proteins, cell death during the early stages of infection remains active. Because *ATL9* is only significantly induced at very early time points during infection, the levels of FBS1, PCC1 and PDF1.2 will begin to accumulate over time as the infection progresses. Our callose staining results show that *atl9*, *fbs1*, *pcc1*, and *pdf1.2* all exhibit faster callose deposition compared to Col-0 during early infection (Figure 7). This indicates that *ATL9*, *FBS1*, *PCC1* and *PDF1.2* may also be involved in impeding callose deposition at sites of potential hyphal penetration. Previous studies have shown that callose and callose synthase are negative regulators of the SA pathway. Therefore, we suggest that *ATL9* might be influencing the SA, JA and ethylene pathways through the degradation of *FBS1, PCC1, and PDF1.2*. Trypan blue staining of powdery mildew infected leaf tissue showed that mutants of *fbs1*, *pcc1* and *pdf1.2* exhibit slightly more cell death than Col-0 and the *atl9* mutant (Figure 8). This indicates that *FBS1*, *PCC1* and *PDF1.2* may also be involved in the regulation of cell death during fungal infection. Combining the observed results from the callose and trypan blue staining, we predict that ATL9, FBS1, PCC1, and PDF1.2 appear to be important in two distinct phases of the fungal infection process: first during the initial phase of infection and second during the propagation of the infection. During the initial phases of infection, *FBS1*, *PCC1* and *PDF1.2* are significantly induced after increased expression of *ATL9*. Cell death during the initial infection period will be activated once FBS1, PCC1 and PDF1.2 are degraded by ATL9 via the proteasome. However, since *ATL9* is only highly expressed at very early points during infection, FBS1, PCC1 and PDF1.2 will begin to re-accumulate in plant cells as the infection progresses and then begin to inhibit callose deposition. Inhibition of callose deposition will trigger cell death via activation of the SA pathway [43,44]. It is of interest to note that *PDF1.2* is continuously induced in *Col-0* plants after infection, suggesting that induction of *PDF1.2* may be involved not only in the early defense response, but also in later downstream responses involved in the persistence or propagation of the defense response. In summary, our study has shown that the E3 ubiquitin ligase ATL9 can affect the expression of defense related genes, localized cell death, and callose deposition in response to fungal infection. These data indicate that ATL9 may have more broad reaching effects in regulating cellular activities during fungal infection.

## 4. Materials and Methods

### 4.1. Biological Materials

All experiments were conducted with *Arabidopsis thaliana* ecotype Col-0. T-DNA insertional mutants were obtained from the Arabidopsis Biological Resource Center (ABRC, Ohio State University, Columbus, USA). To identify homozygous T-DNA mutants, PCR reactions were performed using the following primers: 

*pdf1.2* (Salk_063966 exon):

LP 5′-AGATAAGATGCACCGTCGATG-3′

RP 5′-ATTTGTTCGACGATGACGAAG-3′

BP 5′-GCGTGGACCGCTTGCTGCAACT-3′

*pcc1* (Sail_705_G06 promoter):

LP 5′-TTGATGATGCACCAAACTTTG-3′

RP 5′-TATTGGCAGTTGTATCCAGGG-3′

BP 5′-TAGCATCTGAATTTCATAACCAATCTCGATACAC-3′

*fbs1* (Sail_535_D08 promoter):

LP 5′-TCTCTGTTTCACGGTGTCTCC-3′

RP 5′-TTGTGGGAAATGAACAAAAGC-3′

BP 5′-TAGCATCTGAATTTCATAACCAATCTCGATACAC-3′

Mutants of *atl9* and *ATL9* overexpression line were verified from our previous research [11,12]. Plants were grown under controlled conditions in a growth chamber at 22 °C day/19 °C night with 16 h of light per 24 h and 50% relative humidity. The fungal pathogen used in this work was *Golovinomyces cichoracearum UCSC1*. *G. cichoracearum* was cultured on cucumber (*Cucumis sativus*) and maintained at 22 °C day/19 °C night with 16 h of light per 24 h and 85% relative humidity. Fungal inoculums were prepared, and inoculations performed as described [45]. 

### 4.2. Disease Assessments 

Powdery mildew inoculations and disease assessments were carried out as described [45]. *Arabidopsis* seeds were planted in soil and grown in a growth chamber, as described previously. Col-0 and salicylic-acid-impaired *NahG* plants were planted as controls. After four weeks, plants were inoculated with powdery mildew and placed in a chamber under the same temperature and light conditions but with 85% relative humidity. Disease development was assessed in a qualitative manner by tracking the appearance of powdery mildew symptoms on inoculated leaves over a period of 10 days post inoculation. Leaves at 6 days post inoculation (dpi) were harvested and infected leaf tissue was collected and weighed. Infected leaf tissue (500 mg) was placed in a 15 mL falcon tube containing 5 mL of distilled water. Fungal spores were then suspended in the distilled water by vortexing leaf tissue for approximately 1 minute. The dH_2_O/spore supernatant was then decanted to a fresh tube for quantification. Spores were then quantified using a counting chamber (20 μL of supernatant/chamber), and the results were standardized as spores per mg of leaf tissue [46]. Each experiment was repeated three times and results were analyzed using a student’s *t*-test.

### 4.3. Yeast Two-Hybrid Screening for Protein-Protein Interaction

The Matchmaker GAL4-based Yeast two-hybrid experiments were performed as described by the manufacturer (Clontech, Mountain View, CA, USA). Full-length cDNA of *ATL9* (At2g35000) was amplified from *Arabidopsis* genomic DNA using primers containing NdeI and PstI restriction enzyme sites. Primer sequences are as follows:

5′-GGAATTCATATGCAAACCAGACGTTGACTCTCTCTA-3′(Forward); 

5′-AACTGCAGAAACGCAAATAAGAGGCATGACAA-3′ (Reverse). 

PCR products were recombined with pGBKT7 by restriction enzyme digestion to generate the binding domain construct (pGBKT7-ATL9). After confirming the sequence correctness and direction, the recombinant construct was then transformed into yeast strain Y2H Gold as bait. Full-length cDNA of *PDF1.2*, *PCC1* and *FBS1* were then cloned into pGADT7 to generate the activation domain as described above (pGADT7-PDF1.2, pGADT7-PCC1, and pGADT7-FBS1). The following gene specific primers were used for cloning: 

*PDF1.2* (At5g44420): 

5′-CCGGAATTCACACAACACATACATCTATACATTGA-3′ (Forward)

5′-CGCGGATCCACACACGATTTAGCACCAAAGATT-3′ (Reverse) 

*PCC1* (At3g22231): 

5′-CCGGAATTCCAAATCTCACATCCTCACTCCTCA-3′ (Forward)

5′-CGCGGATCCAACGACTTCTGTCTCATCATGCT-3′ (Reverse) 

*FBS1* (At1g61340): 

5′-CCGGAATTCTCTCAGCTTTTGTTCATTTCCCAC-3′ (Forward)

5′-CGCGGATCCTCCCATAGAGTTTGTTTGTGACCT-3′ (Reverse) 

The above recombinant constructs were then transformed into yeast strain Y187 as prey using the small-scale transformation procedure as described by the manufacturer (Clontech, Mountain View, CA, USA (accessed on 18 March,2018); https://www.takarabio.com) (accessed on 18 March 2018).

Y187 preys (in pGADT7) and Y2H Golds bait(pGBKT7-ATL9) cultures were mixed and mated overnight (20–24 h) with shaking at 200 rpm at 30 °C. Prey pGADT7-T and bait pGBKT7-53 was used as a positive control and prey pGADT7-T and bait pGBKT7-Lam was used as a negative control. For false positive interactions, empty pGBKT7 was mixed with preys (in pGADT7) and empty pGADT7 was mixed with baits (pGBKT7-ATL9). After a 20-h incubation, the mated yeast strains were spread on selective plates with SD/-Leu/-Trp and then SD/-His/-Leu/-Trp medium. The positive colonies were picked and then cultured at SD/-Ade/-His/-Leu/-Trp/X-α-gal plates. Plates were incubated for 3–5 days at 30 °C to screen for blue positive colonies, indicating protein-protein interactions (Clontech). 

### 4.4. Bimolecular Fluorescence Complementation Assays

To confirm the results of the Y2H assay, bimolecular fluorescence complementation (BiFC) assays were also conducted with constructs of pDEST-VYCE and pDEST-VYNE [47]. *ATL9*, *PDF1.2*, *PCC1* and *FBS1* were amplified by PCR as described above. An adenine (A) overhang was added to the amplified PCR products using Taq polymerase and then the constructs were cloned into the PCR8/TOPO/GW entry vector (Invitrogen, Carlsbad, CA). pDEST-VYCE-*ATL9*, pDEST-VYNE-*PDF1.2*, pDEST-VYNE-*FBS1*, and pDEST-VYNE-*PCC1* were then generated by LR reaction (Invitrogen, Carlsbad, CA, USA). All constructs were transformed into *Agrobacterium tumefaciens* strain GV3101 using the freeze/thaw method. Cultured *A. tumefaciens* cells were harvested at OD_600_ = 1.0 and then were re-suspended in fresh infiltration media (containing 10 mM MES/KOH (pH 5.6), 10 mM MgCl_2_, and 150 µM acetosyringone) to yield a final OD_600_ of ~0.5. Re-suspended cultures were incubated for another 2 h at 28 °C with moderate shaking (75 rpm) in darkness [21]. The BiFC partner strains were co-infiltrated into *Nicotiana benthamiana* leaves using a blunt syringe and infiltration spots were outlined using a sharpie marker. Thirty-six hours post-infiltration fluorescence was imaged in the tobacco leaves using a confocal laser-scanning microscope (Nikon Eclipse Ti2, Tokyo, Japan). For the confocal laser-scanning microscopy, two filters were used, the FITC filter (Excitation: 460–500 nm; Emission: 510–560 nm) and the TRITC filter (Excitation: 530–560 nm; Emission: 590–650 nm) (Nikon, Tokyo, Japan). Images were generated and merged using NIS-Elements software Version 4.60.00 (Nikon, Tokyo, Japan).

### 4.5. Analysis of Protein Degradation in Tobacco Leaves

For in vivo detection of ubiquitinated PDF1.2, PCC1, and FBS1, C-terminal GFP-fusion constructs of each gene *(PDF1.2*-*GFP*, *PCC1*-*GFP*, *FBS1*-*GFP*) were produced in the same way as the BiFC constructs, except that the destination vector used was pMDC43 (ABRC). *Agrobacterium tumefaciens* strain GV3101 containing 35S:*ATL9* and one of the recombinant GFP constructs was then co-infiltrated into tobacco leaves [21]. After 36 h incubation, leaves were treated with either the protein synthesis inhibitor cycloheximide (CHX) or CHX plus the proteasome inhibitor MG132, by immersing the infiltrated leaf in MS medium containing either 100 µM CHX or 100 µM CHX plus 100 µM MG132 for 1 h at room temperature [48]. Fluorescence was then monitored using a confocal laser-scanning microscope (Nikon Eclipse Ti2, Tokyo, Japan). For the confocal laser-scanning microscope, FITC filter (Excitation: 460–500 nm; Emission: 510–560 nm) (Nikon, Tokyo, Japan) was used to detect the GFP signal. Images were generated and merged using NIS-Elements software Version 4.60.00 (Nikon, Tokyo, Japan).

Protein degradation was also verified by immunoblotting. Co-transformed tobacco leaf samples were flash frozen using liquid nitrogen and ground into a powder after 0 and 6 h of CHX treatment. Total protein was then extracted using Tris extraction buffer [47]. Proteins were then separated by SDS-PAGE, blotted, transferred to PVDF transfer membrane and probed using polyclonal anti-GFP (Invitrogen, Carlsbad, CA, USA). Results were visualized using chemiluminescence (ThermoFisher, Waltham, MA, USA). 

### 4.6. Gene Expression Pattern by qRT-PCR

21-day old seedlings were infected with the powdery mildew pathogen (*G. cichoracearum)* as previous described [46] and six leaves from six seedlings/genotype were randomly chosen and collected at 0-, 1.5-, 2-, 4-, 8-, 16- and 24-hours post-infection. Total RNA was isolated from infected leaf tissue using the RNeasy Plant Mini Kit (Qiagen, Germantown, MD, USA) and then treated with RQ1 DNase I (Promega, Madison, WI, USA) according to the manufacturer’s protocol. First-strand cDNA was synthesized using the High Capacity cDNA Reverse Transcription Kit (Applied Biosystems, Foster City, CA, USA). The qRT-PCR was performed using iTaq^TM^ Universal SYBR Green Supermix (BIO-RAD, Hercules, CA, USA) with reactions performed at a final volume of 10 µL per well and using the cycle protocol recommended by the manufacturer (95 °C for 2 min, 40 cycles of 95 °C for 15 s and 60 °C for 30 s, 55–95 °C for 10 s). Samples were then run using a StepOne Real-Time PCR System (Applied Biosystems, Foster City, CA, USA). Gene specific primers were designed using the Geneious program (Biomatters, Newark, NJ, USA). The following gene specific primers were used for qRT-PCR; 

ATL9:

5′-TGGAAGAACGTGAAAACTGTGC-3′ (Forward)

5′-GGATCCCACACAACCTTCTCTT-3′ (Reverse)

PDF1.2: 

5′-TGTTCTCTTTGCTGCTTTCGAC-3′ (Forward) 

5′-TGCTGGGAAGACATAGTTGCAT-3′ (Reverse) 

PCC1: 

5′-CGTGCAAAAACCTTCAGAGACT-3′ (Forward) 

5′-AACTCATTATGGCTTCGGGGTT-3′ (Reverse) 

FBS1: 

5′-CGCGTATAGTACACCTCGGAAA-3′ (Forward)

5′-GAATAGAGCCACTGAGACTCCG-3′ (Reverse) 

Beta-Actin: 

5′-CATGCCATCCTCCGTCTTGA-3′ (Forward) 

5′-AGCAGCTTCCATTCCCACAA-3′ (Reverse) 

Changes in transcript levels were determined using the 2^−∆∆CT^ method. Three independent biological replicates were performed for each experiment. 

### 4.7. Trypan Blue Staining for Cell Death

Cell death was visualized in leaves via trypan blue staining [49,50]. Leaf samples were collected with scissors and the cut leaf tissue was then immersed in 2 mL of fresh trypan blue solution (lactic acid: phenol: glycerol: dH_2_O = 1:1:1:1, trypan blue with final concentration of 10 mg/mL) in a culture plate for 30 min to 1 h at room temperature. Trypan blue solution was then replaced by 2 mL of 98–100% ethanol and the culture plate was allowed to sit on the bench for 2 min. The ethanol solution was then changed several times until the leaf tissue became colorless. 60% glycerol was then used to replace the ethanol and leaf samples were mounted on a microscope slide for observation. 

### 4.8. Aniline Blue Staining for Callose Deposition

Callose deposition was observed using aniline blue staining based on standard protocols [47]. Briefly, infected leaves were harvested, placed into a culture plate and distained in lactophenol solution for 2 days at room temperature. The leaf tissue was then washed in 50% ethanol. Cleared leaf samples were then incubated in an aniline blue (MACRON, Center Valley, PA, USA) solution for 2 days in the dark at room temperature. The aniline blue staining solution was discarded and aniline blue stained leaves were then mounted onto a glass slide and observed using a confocal microscope with UV-light excitation. The number of callose deposits per leaf were counted using ImageJ software as described [51,52]. 

## Figures and Tables

**Figure 1 pathogens-11-00068-f001:**
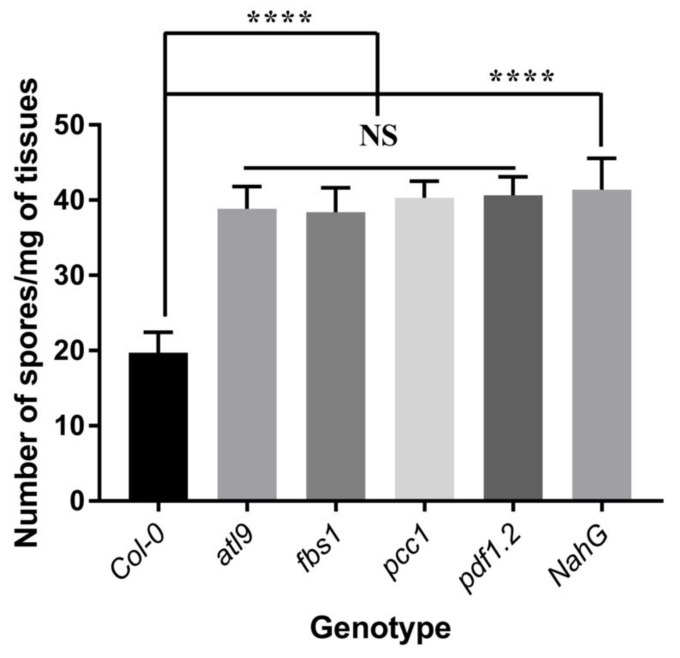
Quantification of *G. cichoracearum* infection on Columbia (Col-0) wild type and *atl9*, *fbs1*, *pcc1* and *pdf1.2* T-DNA insertional mutants. The number of spores per milligram of leaf tissue (spores/mg) was counted 6 dpi. Mutants in the four genes are significantly more susceptible to *G. cichoracearum* than Col-0 wild-type plants. The SA-deficient NahG transgenic plant was used as a positive control and displayed a hypersusceptible phenotype. One-way ANOVA was used to analyze the data. The error bar indicates means ± the SD of three independent biological replicates. **** indicates *p* < 0.0001, and NS indicates not significant. The black solid zig-zag line indicates that the significant differences are present between datasets.

**Figure 2 pathogens-11-00068-f002:**
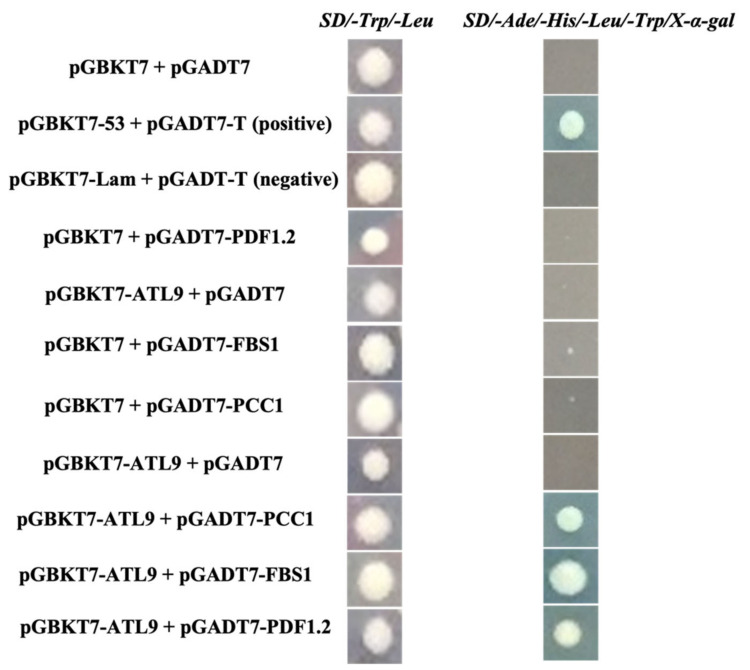
Yeast two-hybrid assay. The Matchmaker GAL4-Yeast two-hybrid experiments were performed as described by the manufacturer (Clontech, Mountain View, CA, USA). Positive results are indicated by blue yeast colonies and were observed in the pGBKT7-ATL9/pGADT7-FBS1, pGBKT7-ATL9/pGADT7-PCC1, and pGBKT7-ATL9/pGADT7-PDF1.2 combinations. pGBKT7-53/pGADT7-T and pGBKT7-Lam/pGADT7-T were used as positive and negative controls respectively. Experiments were repeated three times with independent samples to confirm the results.

**Figure 3 pathogens-11-00068-f003:**
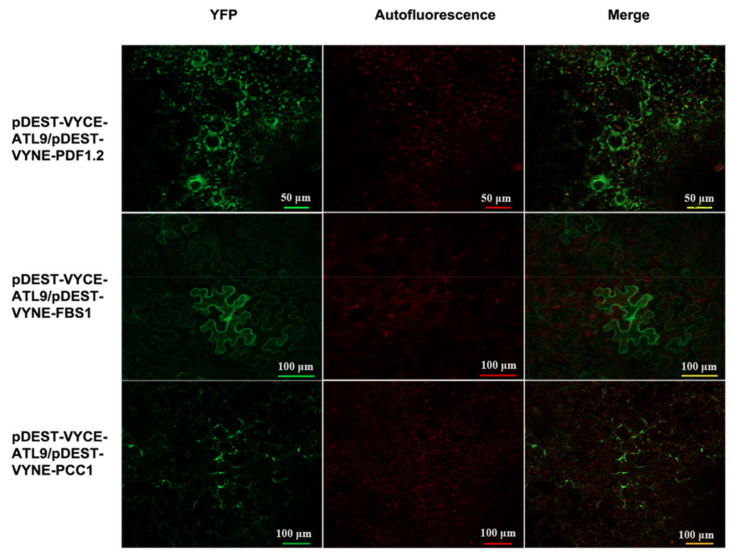
Bimolecular fluorescence complementation assay. pDEST-VYCE-ATL9 was transiently co-expressed with pDEST-VYNE-FBS1, pDEST-VYNE-PCC1, and pDEST-VYNE-PDF1.2 in *N. benthamiana* leaves. YFP fluorescence was observed using confocal microscopy. Yellow fluorescent emission indicated a positive interaction. FITC channel (Excitation: 460–500 nm; Emission: 510–560 nm) was used to detect YFP signal and TRITC (Excitation: 530–560 nm; Emission: 590–650 nm) filters were used to indicate autofluorescence.

**Figure 4 pathogens-11-00068-f004:**
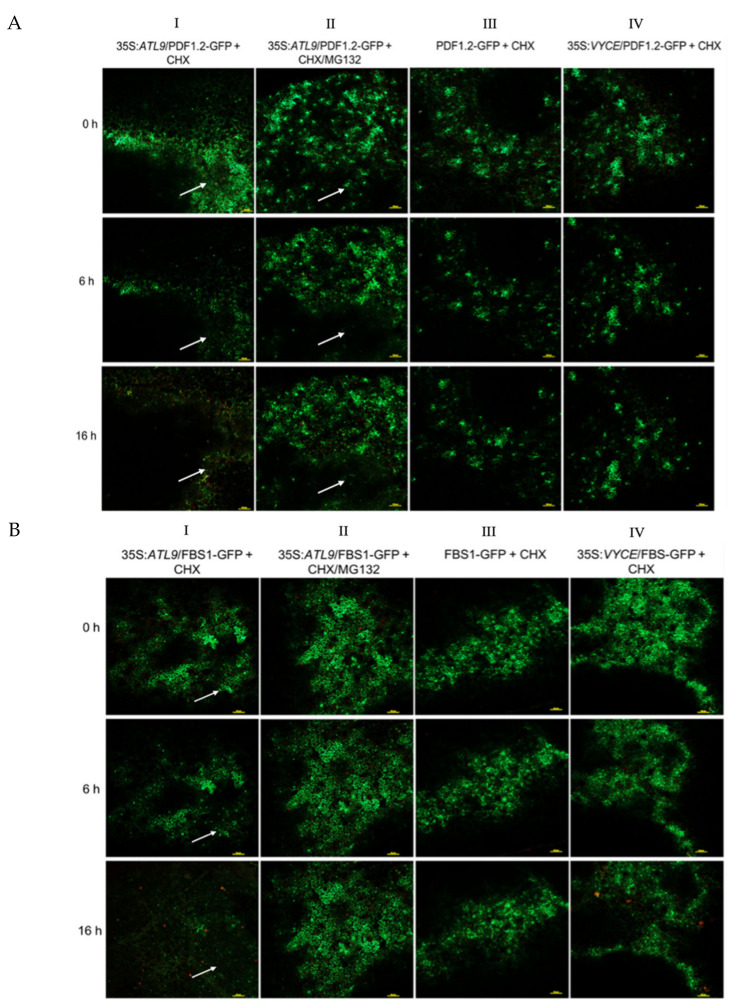
In vivo degradation assay. (**A**) 35S:ATL9 was transiently co-expressed with PDF1.2-GFP in *N. benthamiana* leaves. After 36 h incubation, leaves were treated with either 100 µM CHX or 100 µM CHX plus 100 µM MG132 before observation and then fluorescence emission was monitored with the confocal microscope. Fluorescent emission of PDF1.2-GFP decreased quickly when it was co-infiltrated with 35S:ATL9 (column I). Inhibition of the 26S proteasome with MG132 significantly reduces the amount of GFP degradation, indicating that PDF1.2 is degraded via the proteasome (column II). Fluorescent emission of PDF1.2-GFP alone was persistent over the course of the experiment (column III). PDF1.2:GFP combined with an empty vector shows little to no change in degradation of the signal(column IV). The vector alone that was used for cloning ATL9 did not cause any visible degradation of PDF1.2-GFP (Appendix A). (**B**) 35S:ATL9 was transiently co-expressed with FBS1-GFP in *N. benthamiana*. (**C**) 35S:ATL9 was transiently co-expressed with PCC1-GFP in *N. benthamiana*. The FITC channel was used to detect GFP signal and the yellow scale bar indicates 100 µm. The white arrows indicate the areas where the protein is significantly degraded.

**Figure 5 pathogens-11-00068-f005:**
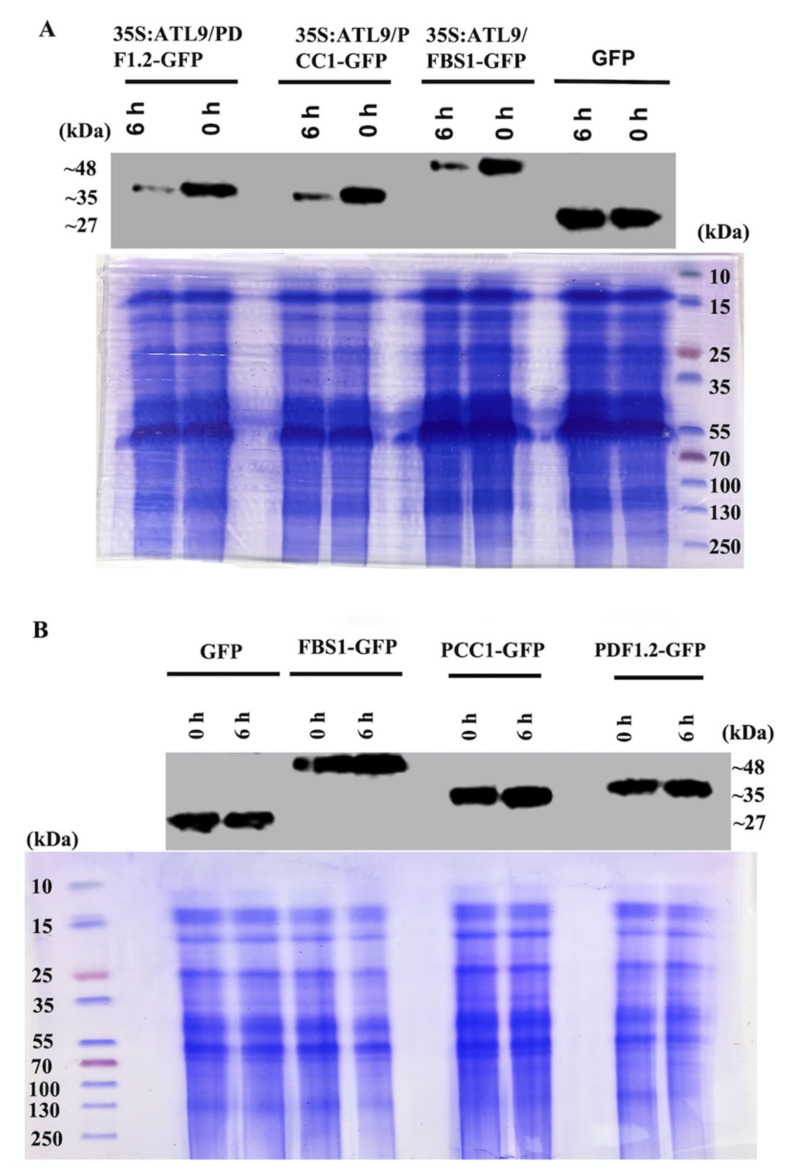
Immunoblotting of in vivo degradation assay. (**A**) PDF1.2-GFP, PCC1-GFP, and FBS1-GFP are co-infiltrated with 35S:ATL9. (**B**) GFP, PDF1.2-GFP, PCC1-GFP, and FBS1-GFP are infiltrated alone. Proteins were extracted from co-infiltrated tobacco leaves at 0 and 6 h after CHX treatment. Proteins were separated by SDS-PAGE and immunoblotted with anti-GFP. Samples transformed by GFP-fused vector alone were used as control. The quantity of PDF1.2-GFP, FBS1-GFP and PCC1-GFP decreased when these proteins were co-infiltrated with 35S:ATL9.

**Figure 6 pathogens-11-00068-f006:**
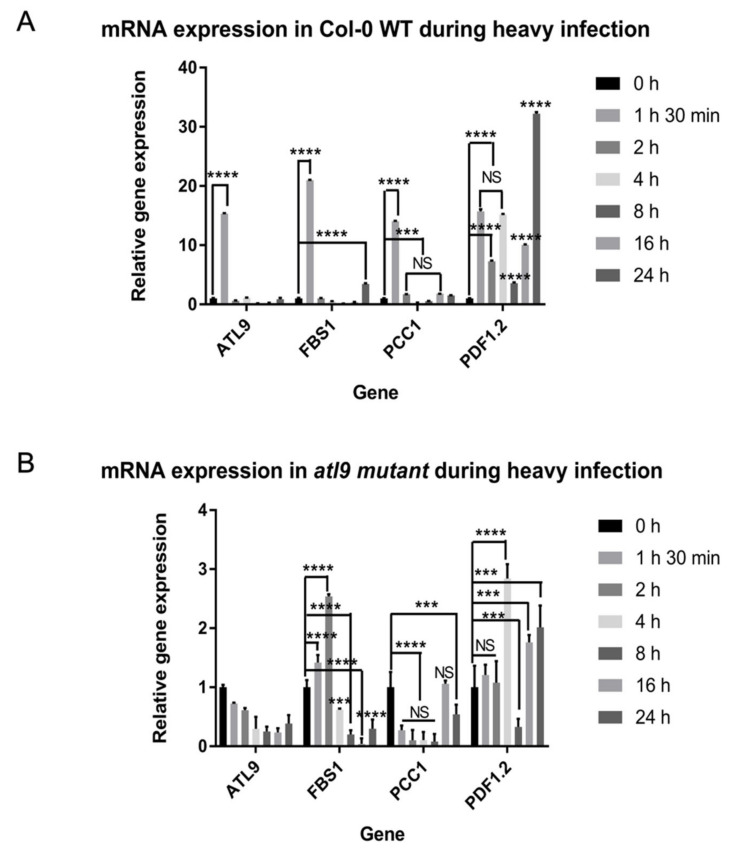
Expression of *ATL9*, *FBS1*, *PCC1* and *PDF1.2* in Columbia wild type and *atl9* plants after infection with *Golovinomyces cichoracearum.* (**A**) mRNA expression of *PDF1.2*, *FBS1*, *ATL9*, and *PCC1* in Col-0 wild type plants after heavy infection with powdery mildew. (**B**) mRNA expression of *PDF1.2*, *FBS1*, *ATL9*, and *PCC1* in the *atl9* mutant after heavy infection with powdery mildew. Beta-actin was used as an endogenous control. Please note change in y-axis scales between (**A**) and (**B**). Asterisks indicate statistically significant differences between the samples treated and untreated, according to One-way ANOVA analysis and multiple comparison post-Tukey’s test. The experiments were repeated three times with independent biological samples. **** indicates *p* < 0.0001, *** indicates *p* < 0.001, and NS indicates not significant. The black solid zig-zag line indicates that the significant differences are present between datasets.

**Figure 7 pathogens-11-00068-f007:**
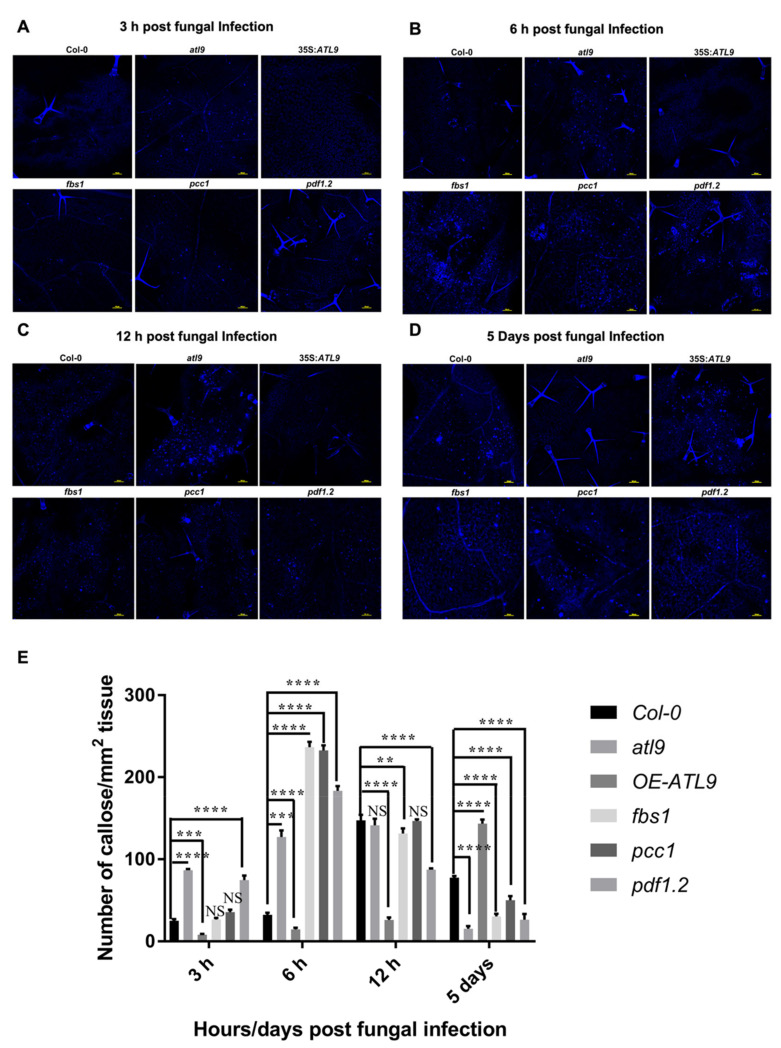
Callose staining of leaves after infection for 3, 6 and 12 h or 5 days post infection with powdery mildew. Plants infected with powdery mildew after (**A**) 3 h (**B**) 6 h (**C**) 12 h and (**D**) 5 days had leaves harvested and then stained for callose deposition with aniline blue. After staining, samples were observed using confocal microscopy. (**E**) Quantitative assessment of the number of callose deposits per mm^2^ leaf tissue. Asterisks indicate statistically significant differences between the samples treated and untreated, according to One-way ANOVA analysis and multiple comparison post-Tukey’s test. **** indicates *p* < 0.0001, *** indicates *p* < 0.001, ** *p* < 0.01 and NS indicates not significant. The black solid zig-zag line indicates that the significant differences are present between datasets. Yellow scale bars equal to 100 μm.

**Figure 8 pathogens-11-00068-f008:**
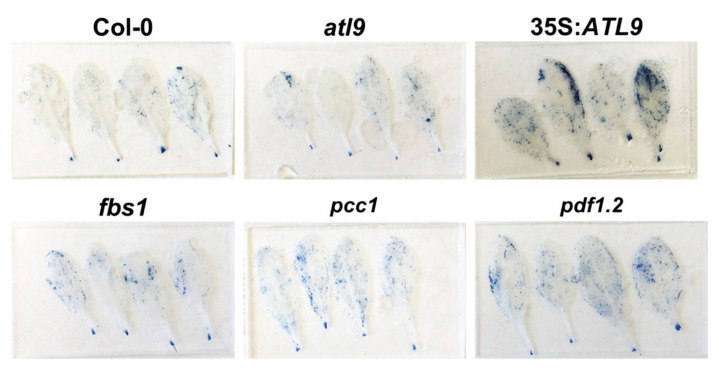
Cell death in leaves indicated by trypan blue staining. Plants were infected with powdery mildew for 5 days and leaf tissues were collected and stained with trypan blue. Results show that cell death in leaf epidermal cells is largely increased in *35S:ATL9* plants and slightly increased in *fbs1*, *pcc1*, and *pdf1.2* mutants.

**Figure 9 pathogens-11-00068-f009:**
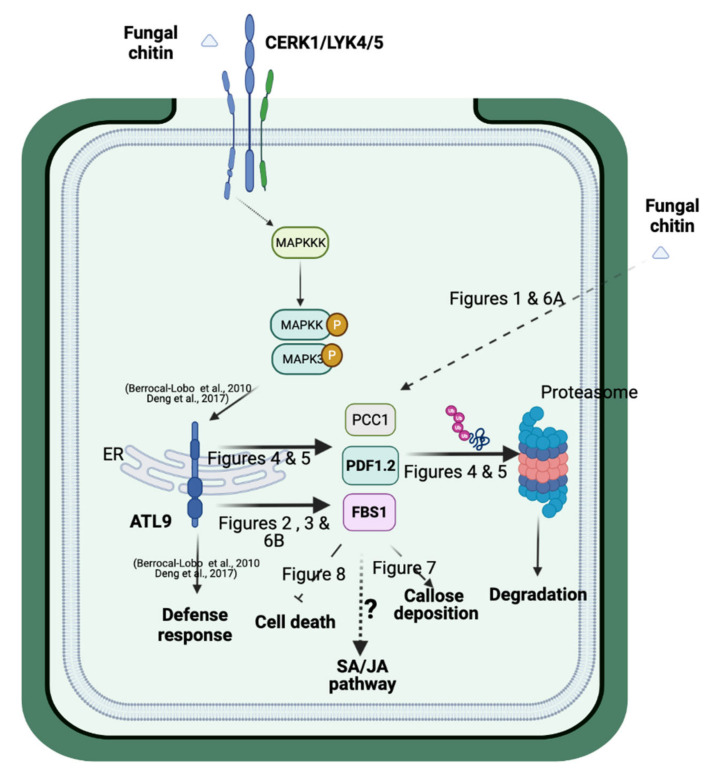
Schematic model of ATL9′s function in plant defense response. Representation of the possible role of ATL9 during plant defense responses to fungi [11,12]. Black arrows indicate positive regulation, whereas black end-blocked lines indicate negative regulation. Dashed lines indicate possible interactions with pathways or molecules. The figure was created with BioRender.com with license agreement number FV2337XIJK (accessed on 18 October 2021).

## Data Availability

Data is contained within the article or Appendix A.

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
