# Peer review of "The E3 Ubiquitin Ligase ATL9 Affects Expression of Defense Related Genes, Cell Death and Callose Deposition in Response to Fungal Infection"

_pathogens, 2022, doi:10.3390/pathogens11010068_

Round 1
Reviewer 1 Report
The manuscript studies the participation of ATL9 ubiquitin ligase and PDF1.2, PCC1 and FBS1 proteins in Arabidopsis response to the fungal pathogen Golovinomyces cichoracearum and their role in different defense-related processes such as callose deposition or cell death.
In my view, the manuscript shows difficult readability. I found many incomplete sentences, extra words, missing or innacurate explanations, missing references, redundant language, sentence connectors not properly used or contradictory sentences. Further, some results are incorrectly included in discussion section. I suggest the authors to make a profound revision of writing, avoid redundant explanations and show more organized sections and clearer conclusions.
I also encourage the authors to revise the literature to avoid incorrect and inaccurate statements. More detailed background of the proteins studied and their implication in plant defense pathways is needed to better integrate the results obtained. The results are not properly discussed and there are plenty of statements which are not supported by the data.
-Incorrect/Innacurate sentences:
79.“Plant defensin 1.2 (PDF1.2), has a myriad of functions, including response to insects, the induction of salicylic acid (SA), jasmonic acid (JA) [16]” PDF1.2 is a final product of JA/ET pathway. Please check your references.
419-423. “This indicates that ATL9, FBS1, PCC1 and PDF1.2 may also be involved in impeding callose deposition” “FBS1, PCC1 and PDF1.2 can indirectly activate the SA pathway”. 413. “FBS1, PCC1 and PDF1.2, which inhibit cell death during early infection are also upregulated” 426. “This indicates that FBS1, PCC1 and 426 PDF1.2 may also inhibit cell death either directly or indirectly” These statements are not supported by the data. PDF is an example of final product of JA/ET pathways that is secreted from the cell, so the data does not lead to speculate that the protein is impeding callose deposition or inhibiting cell death by itself. The absence of these proteins in the mutant plants of this study could be producing a deregulation of defense signalling pathways that lead to the phenotypes observed.
How do you explain that the T-DNA insertional mutant atl9 affects decreasing transcriptional levels of FBS1, PCC1 and PDF1.2 (Fig. 5) and at the same time ATL9 promotes these proteins’ degradation? (Fig. 3). This is not properly discussed.
32. The zigzag model with branches is difficult to understand the way it is explained.
199. “Since atl9, pdf1.2, fbs1 and pcc1 mutants are more susceptible to G. cichoracearum than wild-type plants” reference missing/ This statement comes from Fig 8?
105-106. “Since mutations in PDF1.2, PCC1, and FBS1 result in a significantly altered plant defense phenotype when challenged with powdery mildew”. Reference missing. Explain what type of alteration.
36. “Pathogens that are successfully colonized can deliver virulence factors” Pathogens are not colonized. Plant is the one colonized.
103. “ATL9, which is highly induced by fungal pathogens”. ATL9 is induced (in the plant) in response to fungal pathogens.
318 “proteins PDF1.2, PCC1 and FBS1 directly interact with and are attenuated by ATL9”
154. Post infiltration/ after coinfiltration/transformation…, instead of incubation.
185. Degradation of the genes not GFP degradation.
359-364. These statements are confusing, seem contradictory, and are not well connected.
334. “These results indicate that FBS1, PCC1 and 333 PDF1.2 may play a critical role in plant defense against powdery mildew and are potential protein targets of ATL9” this conclusion can not be extracted from Fig 8.
287. “The callose staining and expression pattern of FBS1, PCC1 and PDF1.2 indicate that both SA- and JA-signaling pathways could be associated with the expression of ATL9”. Please explain
202. "Heavily inoculated” Fig. 5 “Heavy infection”. The word heavy is not precise enough.
-Figures:
Fig 1. Next to the images, the name of the genes are indicated without the vector and when the vector is empty the name of the vector is indicated. Please standardize.
Fig. 2. Nothing can be observed in pDEST-VYCE-ATL9/pDEST-VYNE-PDF1.2 and in pDEST-VYCE-ATL9/pDEST-VYNE-PCC1 in TRITC channel, so the integrity of the cells can not be determined. Similarly, it is difficult to predict the localization of the green signal in these two co-infiltrations. Pdf2.1, PCC1 and FBS1 are localized in different plant cell locations than ATL9. How do you explain when and where they interact?
Besides, how would you explain that PDF1.2 is ubiquitinated and degraded in the proteasome (figure 3) being a secreted protein?
Figure 2. Indicating in the figure FITC, TRITC above the images instead of YFP, autofluorescence can be confusing since it seems the signal belongs to FITC, TRITC.
Figure 4. A Ponceau staining of the western blot is often used as a more accurate loading control. Pannels A and B are not cited in figure caption.
-Issues with sections:
Lines 82-94 is a summary of results included in introduction section.
325-327 results included in discussion. Figure 8 are also included in discussion section, and should be placed in results section.
406-407 should be in materials and methods section
Please include in materials and methods how many plants are included in a biological replicate or if technical replicates are performed 549-578.
440 In discussion section, I suggest to focus in connecting the results with the literature, leading to specific conclusions. There is too much explanation of results that have been already explained. A final conclusion is missing in the end of discussion.
-Issues with the use of connectors:
30. However connector is not used properly, sounds weird.
373. in addition (same as above)
359-363. Revise the use of however and in addition connectors.
-Extra words:
42. Within cells
44. other
-Typos:
73. treatments
75. Involved in
360. Involved in
325 extra . (period)
Lines 40, 76, 104, 208, 396. There are extra spaces
Fig 6. Hours/days post infection instead of hours/days of infection
-Missing/incorrect references
199-202 reference missing
321. Incomplete references.
-Incomplete sentences:
113. “As shown in Figure 1, when ATL9 bait (pGBKT7-ATL9) was combined with PDF1.2, PCC1, and FBS1 in the prey vector”.
254. “While very little callose deposition was detected in 35S:ATL9, as shown in Figure 6A”.
324. “To confirm if PDF1.2, PCC1 and FBS1 are involved with defense against fungal infection”.
-Redundant language:
255-266 callose deposition. Results are too descriptive and very little conclusions, it is hard to read.
132. co-infiltrated into Nicotiana benthamiana leaves by infiltration
335 “The spore counting assay showed that mutants of pdf1.2, pcc1 and 335 fbs1 displayed a susceptible phenotype to fungal infection”. Already said in 330.
426. “This indicates that FBS1, PCC1 and 426 PDF1.2 may also inhibit cell death either directly or indirectly”. Already said in 413.
Author Response
Dear Editor and Reviewer,
We appreciate the time that you dedicated to providing feedback on our manuscript and are grateful for the insightful comments and valuable improvements to our paper. We have made changes based on the reviewer’s comments. These changes are highlighted within the manuscript. Please see comments below for a point-by-point response to the reviewer comments.
- “Plant defensin 1.2 (PDF1.2), has a myriad of functions, including response to insects, the induction of salicylic acid (SA), jasmonic acid (JA) [16]” PDF1.2 is a final product of JA/ET pathway. Please check your references.
Page 2 line 92: We have updated the references. We acknowledge that PDF1.2 encodes an ethylene- and jasmonate-responsive plant defensin. However, PDF1.2 was also found to be involved in the plant response to insects and pathogens. Additionally, jasmonate and salicylic acid can act synergistically to enhance the expression of this gene. We deleted the phrase ‘the induction of salicylic acid’ in the manuscript, since PDF1.2 mRNA levels are not responsive to SA treatment.
- 419-423. “This indicates that ATL9, FBS1, PCC1 and 2 may also be involved in impeding callose deposition” “FBS1, PCC1 and PDF1.2 can indirectly activate the SA pathway”. 413. “FBS1, PCC1 and PDF1.2, which inhibit cell death during early infection are also upregulated” 426. “This indicates that FBS1, PCC1 and 426 PDF1.2 may also inhibit cell death either directly or indirectly” These statements are not supported by the data.
- PDF is an example of final product of JA/ET pathways that is secreted from the cell, so the data does not lead to speculate that the protein is impeding callose deposition or inhibiting cell death by itself. The absence of these proteins in the mutant plants of this study could be producing a deregulation of defense signaling pathways that lead to the phenotypes observed.
Thank you for your comments. Our callose staining and cell death results show that atl9, fbs1, pcc1, and pdf1.2 all exhibit faster callose deposition or increased cell death compared to Col-0 during early infection. Thus, in our proposed model, we suggested that ATL9, FBS1, PCC1 and PDF1.2 may be involved in callose deposition and/or cell death in fungal defense. We agree with the reviewer that our data does not support that FBS1, PCC1 and PDF1.2 are directly involved with callose deposition and cell death by themselves, a point which we did not conclude in our proposed model. It may be possible that the absence of these proteins in the mutants used in this study produce a deregulation of defense signaling pathways that lead to the phenotypes observed as is the reviewer’s speculation. Our data only suggested that FBS1, PCC1 and PDF1.2 are involved in some way with the regulation of cell death and/or callose deposition during fungal infection. Further experiments would be necessary to determine if ATL9 degrades these proteins and to show that these proteins influence callose deposition, as we mentioned in the discussion (page 12 line 348). To avoid confusion, we made the following changes.
Page 14 Figure 9: We updated Figure 9. The effects of PCC1, PDF1.2, and FBS1 on cell death and callose deposition with dashed lines, indicating unknown regulation.
Line 1312: We modified the ‘also be involved in’ to ‘This indicates that ATL9, FBS1, PCC1 and PDF1.2 may also be involved in the regulation of’.
Line 1331: We agree that the statement that FBS1, PCC1 and PDF1.2 can indirectly activate the SA pathway is an overstatement and not supported by the data. Since previous studies have shown that FBS1, PCC1, and PDF1.2 are involved in all three major defense pathways (SA, JA and ethylene) [14,27,38] we suggested that ATL9 might be involved with the SA, JA and ethylene pathways via FBS1, PCC1, and PDF1.2. We have modified the statement ‘FBS1, PCC1 and PDF1.2 can indirectly activate the SA pathway’ to ‘we suggest that ATL9 might be influencing the SA, JA and ethylene pathways via FBS1, PCC1, and PDF1.2’.
Line 1307: We modified the statement “FBS1, PCC1 and PDF1.2, which inhibit cell death during early infection are also upregulated’ to ‘FBS1, PCC1 and PDF1.2 are also upregulated (Figure 6A)’.
Line 1334: We modified the statement ‘This indicates that FBS1, PCC1 and 426 PDF1.2 may also inhibit cell death either directly or indirectly’ to ‘This indicates that FBS1, PCC1 and PDF1.2 may also be involved in the regulation of cell death during fungal infection.’
- How do you explain that the T-DNA insertional mutant atl9 affects decreasing transcriptional levels of FBS1, PCC1 and PDF1.2 (Fig. 5) and at the same time ATL9 promotes these proteins’ degradation? (Fig. 3). This is not properly discussed.
In this case, we noted that in T-DNA insertional mutants of atl9 there are decreased transcriptional levels of FBS1, PCC1 and PDF1.2 at all the time points tested during fungal infection (Fig. 6B) which suggests that ATL9 is involved in the regulation of the expression of FBS1, PCC1 and PDF1.2 during fungal infection. In the in vivo ubiquitination assay (Figure 4), FBS1 and PDF1.2 are ubiquitinated by ATL9 at 6h and PCC1 is ubiquitinated by ATL9 at 16h. Our data only suggest that ATL9 is able to degrade the FBS1, PCC1, and PDF1.2 via ubiquitination. In addition, our previous study showed that ATL9 is a short-lived protein. The degradation of FBS1, PCC1 and PDF1.2 in reality might be through ATL9 or another ubiquitination pathway. Our thought is that the gene regulation network is not a linear model but time dependent. In Col-0 WT, after fungal infection, we found ATL9 mRNA expression is only upregulated at early time points, while FBS1, PCC1 and PDF1.2 expression varies at different timepoints after fungal infection. The expression of FBS1, PCC1 and PDF1.2 during fungal infection might be also regulated by another signaling network during fungal infection. We acknowledge that there is a missing link between the ubiquitination pattern of ATL9, FBS1, PCC1, and PDF1.2. Thus, we mentioned in the discussion section (page 12 line 830) that a future avenue of study would be to investigate these protein ubiquitination patterns more closely by ATL9.
- The zigzag model with branches is difficult to understand the way it is explained.
Page 1 line 30-44: We reorganized the introduction section for the zigzag model for clarity.
- “Since atl9, pdf1.2, fbs1 and pcc1 mutants are more susceptible to G. cichoracearum than wild-type plants” reference missing/ This statement comes from Fig 8?
Page 8 line 443: We updated the reference and the statement comes from Figure 1-3.
- 105-106. “Since mutations in PDF1.2, PCC1, and FBS1 result in a significantly altered plant defense phenotype when challenged with powdery mildew”. Reference missing. Explain what type of alteration.
Page 3 line 191: The reference was added and the sentence was modified. The sentence “Since mutations in PDF1.2, PCC1, and FBS1 result in a significantly altered plant defense phenotype when challenged with powdery mildew” was modified to “Since mutations in PDF1.2, PCC1, and FBS1 result in a susceptible phenotype when challenged with powdery mildew (Figure 1)”.
- “Pathogens that are successfully colonized can deliver virulence factors” Pathogens are not colonized. Plant is the one colonized.
Page 1 line 37: We modified the sentence ‘Pathogens that are successfully colonized can deliver virulence factors’ to ‘pathogens that successfully colonize host plants can deliver virulence factors (effectors) into plant cells to counteract the effects of plant PTI, referred to as effector-triggered susceptibility (ETS)’.
- 318 “proteins PDF1.2, PCC1 and FBS1 directly interact with and are attenuated by ATL9”
Page 12 line 839: We deleted ‘and are attenuated by’
- Post infiltration/ after coinfiltration/transformation…, instead of incubation.
Page 5 line 320: We modified ‘incubation’ to ‘post infiltration’.
- Degradation of the genes not GFP degradation.
Page 7 line 412: We modified ‘GFP degradation’ to ‘Degradation of the three genes’.
- 359-364. These statements are confusing, seem contradictory, and are not well connected.
Page 11 line 407-413: To avoid confusion, we deleted the sentence and reorganized the paragraph.
- “These results indicate that FBS1, PCC1 and 333 PDF1.2 may play a critical role in plant defense against powdery mildew and are potential protein targets of ATL9” this conclusion can not be extracted from Fig 8.
Page 3 line 180: Our results only showed that mutants of fbs1, pcc1 and pdf1.2 have a susceptible phenotype to powdery mildew infection, which suggests that FBS1, PCC1 and PDF1.2 may play a critical role in plant defense against powdery mildew. Thus, we deleted the conclusion that ‘FBS1, PCC1 and PDF1.2 are potential protein targets of ATL9’.
- “The callose staining and expression pattern of FBS1, PCC1 and PDF1.2 indicate that both SA- and JA-signaling pathways could be associated with the expression of ATL9”. Please explain
Page 11 line 746: To avoid confusion, we modified the sentence ‘The callose staining and expression pattern of FBS1, PCC1 and PDF1.2 indicate that both SA- and JA-signaling pathways could be associated with the expression of ATL9’ to ‘Previous studies showed that FBS1, PCC1, and PDF1.2 are involved in the three major defense pathways mediated by SA, JA and ethylene [14,27]. The expression levels of FBS1, PCC1, and PDF1.2 are reduced in the atl9 mutant during fungal infection, suggesting that ATL9 might be involved in the regulation of these hormonal pathways via interaction with FBS1, PCC1, and PDF1.2’.
- "Heavily inoculated” Fig. 5 “Heavy infection”. The word heavy is not precise enough.
Page 8 line 408 and Page 9 Figure 6: We deleted the word ‘heavily’ and updated figure 6.
- Fig 1. Next to the images, the name of the genes are indicated without the vector and when the vector is empty the name of the vector is indicated. Please standardize.
Page 4 Figure 2: We updated the names of the genes and vectors in Figure 2.
- 2. Nothing can be observed in pDEST-VYCE-ATL9/pDEST-VYNE-PDF1.2 and in pDEST-VYCE-ATL9/pDEST-VYNE-PCC1 in TRITC channel, so the integrity of the cells can not be determined. Similarly, it is difficult to predict the localization of the green signal in these two co-infiltrations. Pdf2.1, PCC1 and FBS1 are localized in different plant cell locations than ATL9. How do you explain when and where they interact? Besides, how would you explain that PDF1.2 is ubiquitinated and degraded in the proteasome (figure 3) being a secreted protein?
Page 5 Figure 3: We performed the BiFC experiments in tobacco leaves. Our FITC channel is designed to detect the YFP signal, which indicates a positive interaction, and the TRITC channel was used to indicate autofluorescence. We don’t have an extra channel on our confocal microscope to simultaneously detect a plasma membrane fluorescence marker to indicate the integrity of the cells. Similarly, we don’t have an extra organelle fluorescence marker to predict the localization of the green signal in these two co-infiltrations. It’s also interesting that PDF1.2 is ubiquitinated and degraded in the proteasome (figure 3) being a secreted protein. We mentioned in discussion section (page 12 line 728) that PDF1.2 was found to localize to the ER bodies, where it can be secreted to apoplastic space after a fungal attack. Although we agree that this is an important consideration, it is beyond our scope and cannot be analyzed in this manuscript. The detailed mechanisms behind ATL9 target ubiquitination largely remain unknown. We mentioned in discussion section (page 12 line 731) that a future avenue of study would be to investigate these protein ubiquitination patterns by ATL9 and to determine the precise amino acid location of ubiquitination in each protein.
- Figure 2. Indicating in the figure FITC, TRITC above the images instead of YFP, autofluorescence can be confusing since it seems the signal belongs to FITC, TRITC.
Page 5 Figure 3: We modified the ‘FITC, and TRITC’ above the images to ‘YFP, and autofluorescence’.
- Figure 4. A Ponceau staining of the western blot is often used as a more accurate loading control. Pannels A and B are not cited in figure caption.
Page 8 Figure 5: We agree with the reviewer that a ponceau staining of the western blot is often used as a more accurate loading control. However, the Coomassie blue staining of the gel is also able to accurately show the loading control. Given the costs and time involved to repeat the whole experiment (culture the tobacco, co-infiltration, western blot, etc.) we have chosen not to repeat this experiment and redo the entire figure. We have updated the panels A and B in figure caption.
- Lines 82-94 is a summary of results included in introduction section.
Page 2 Line 94-105: We appreciate reviewer’s feedback and we have reorganized the introduction section. We have only listed the major points in our study to let the reader know what our main conclusions are and what to expect in the reminder of the manuscript.
- 325-327 results included in discussion. Figure 8 are also included in discussion section, and should be placed in results section.
Page 12 discussion section: Figure 8 was removed from the discussion section and moved to the results section as Figure 1.
- 406-407 should be in materials and methods section
Page 14 line 1251: We removed the figure license agreement number.
- Please include in materials and methods how many plants are included in a biological replicate or if technical replicates are performed 549-578.
Page 17 line 1441: We have updated the materials and methods of the number of plants in a biological replicate in qRT-PCR assay.
- 440 In discussion section, I suggest to focus in connecting the results with the literature, leading to specific conclusions. There is too much explanation of results that have been already explained. A final conclusion is missing in the end of discussion.
We appreciate the reviewer’s suggestion. We reorganized the discussion section and added the final conclusion at the end of discussion.
- However connector is not used properly, sounds weird.
Page 1 line 32, We modified ‘however’ to ‘instead’.
- in addition (same as above)
Page 13 line 508: We modified ‘in addition’ to ‘In the atl9 T-DNA mutant’.
- 359-363. Revise the use of however and in addition connectors.
We reorganized the connector problems throughout the manuscript.
- Within cells 44. other
Page 1 line 42 and Page 2 line 90: We deleted the extra words.
- treatments 75. Involved in 360. Involved in
Page 2 line 86 and 88, and Page 13 line 978: This has been corrected in the manuscript.
- Lines 40, 76, 104, 208, 396. There are extra spaces
This has been corrected in the manuscript.
- Fig 6. Hours/days post infection instead of hours/days of infection
Page 11 figure 7: We modified ‘hours post fungal infection’ to ‘Hours/days post fungal infection.
- 199-202 reference missing
Page 8 line 406-408: This has been corrected in the manuscript.
- Incomplete references.
Page 12 line 720: This has been corrected in the manuscript.
- “As shown in Figure 1, when ATL9 bait (pGBKT7-ATL9) was combined with PDF1.2, PCC1, and FBS1 in the prey vector”.
Page 3 line 200, The sentence was reorganized.
- “While very little callose deposition was detected in 35S:ATL9, as shown in Figure 6A”.
Page 10 line 491: The sentence was reorganized.
- “To confirm if PDF1.2, PCC1 and FBS1 are involved with defense against fungal infection”.
Page 3 line 122: The sentence was reorganized.
- 255-266 callose deposition. Results are too descriptive and very little conclusions, it is hard to read.
Page 10 line 363-390: We reorganized the results section regarding the deposition of callose during fungal infection.
- co-infiltrated into Nicotiana benthamiana leaves by infiltration
Page 4 line 290: e deleted the phrase (by infiltration).
- 335 “The spore counting assay showed that mutants of 2, pcc1 and 335 fbs1 displayed a susceptible phenotype to fungal infection”. Already said in 330.
The sentence was deleted (The spore counting assay showed that mutants of pdf1.2, pcc1 and 335 fbs1 displayed a susceptible phenotype to fungal infection).
- “This indicates that FBS1, PCC1 and 426 PDF1.2 may also inhibit cell death either directly or indirectly”. Already said in 413.
Page 14, line 1319: We deleted sentence (This indicates that FBS1, PCC1 and 426 PDF1.2 may also inhibit cell death either directly or indirectly) and reorganized the paragraph.
Reviewer 2 Report
This study presents a comprehensive mechanistic account of callose deposition mechanism in response to fungal infection in Arabidopsis. It fills several gaps in our understanding of callose deposition occurring as a defense mechanism. The paper, its experimental design, results, analysis of data, statistical testing and conclusions drawn from the experiments do not contain major problems. The manuscript itself is very well written. However, the description made on the confocal laser scanning microscopy (CLSM) is very inadequate. Please elaborate the workflow pertaining to your CLSM procedure in more detail because it forms a significant aspect of the work undertaken by this paper. The yeast two-hybrid screening assay (section 4.3) can also be improved with a little more detail because the amount of detail provided isn't sufficient for someone else to replicate the work described in this section.
Author Response
Dear Reviewer
We appreciate the time that you dedicated to providing feedback on our manuscript and are grateful for the insightful comments and valuable improvements to our paper.
We have made changes based on your comments. Those changes are highlighted within the manuscript. Please see below, in blue, for a point-by-point response to your comments. All page numbers refer to the revised manuscript file with tracked changes.
This study presents a comprehensive mechanistic account of callose deposition mechanism in response to fungal infection in Arabidopsis. It fills several gaps in our understanding of callose deposition occurring as a defense mechanism. The paper, its experimental design, results, analysis of data, statistical testing and conclusions drawn from the experiments do not contain major problems. The manuscript itself is very well written. However, the description made on the confocal laser scanning microscopy (CLSM) is very inadequate. Please elaborate the workflow pertaining to your CLSM procedure in more detail because it forms a significant aspect of the work undertaken by this paper. The yeast two-hybrid screening assay (section 4.3) can also be improved with a little more detail because the amount of detail provided isn't sufficient for someone else to replicate the work described in this section.
Page 16 line 714-721 (method section 4.3): Thank you for pointing this out. We have updated the yeast two hybrid method.
Page 17 line 745-748(methods section 4.4. and 4.5.): We have updated the confocal laser scanning microscopy method.
Reviewer 3 Report
According to the authors, their findings suggest that ubiquitination, cell death, and callose deposition may all work in concert to improve defence responses against fungal pathogens. Overall the manuscript provides valuable results and is well written.
Author Response
Dear Reviewer,
We appreciate the time that you dedicated to providing feedback on our manuscript and are grateful for the insightful comments on and valuable improvements to our paper.
According to the authors, their findings suggest that ubiquitination, cell death, and callose deposition may all work in concert to improve defence responses against fungal pathogens. Overall the manuscript provides valuable results and is well written.
We have updated the references and placed them in the correct order.
Thank you again for your time and consideration.
Round 2
Reviewer 1 Report
Authors greatly improved the readability and clarity of the manuscript.
A few issues:
Line 96. "Mutants of pdf1.2, pcc1 and fbs1 are more susceptible..." (Col-0 is already susceptible)
Line 117. More susceptible
Figure 3. I still can not see anything in at least two autofluorescence images. Cell integrity can also be seen with the bright field channel of the microscope.
Author Response
Dear Editor and Reviewer,
Thank you for your comments. They are appreciated. A point-by-point response to the three comments is below.
Line 96. "Mutants of pdf1.2, pcc1 and fbs1 are more susceptible..." (Col-0 is already susceptible)
Line 117. More susceptible
Line 96 and line 117: We have modified these sentences in the manuscript.
Figure 3. I still can not see anything in at least two autofluorescence images. Cell integrity can also be seen with the bright field channel of the microscope.
Figure 3: We have increased the brightness of the image so that any autofluorescence is visible in all TRITC channel images.
We agree with the reviewer that bright field microscopy would show the surface of the N. benthamiana leaves. However, when this experiment was performed, we did not include bright field images for this BiFC experiment since it is not part of our standard protocol. The variation in depth of the Z-dimension pixels in the confocal microscope when imaging for YFP and Autofluorescence can give blurry and uneven images of the tobacco surface if one tries to use the same settings for a bright field image. Further, the lack of a bright field image of the tobacco leaves does not affect the results or conclusions in this manuscript. Thus given the time involved, we have chosen not to repeat the entire experiment at this time. We have reorganized Figure 3 as suggested by reviewer.